# Wild Plants Used by Tibetans in Burang Town, Characterized by Alpine Desert Meadow, in Southwestern Tibet, China

**Xiaoyong Ding** [1,2,†], **Changan Guo** [1,2,†], **Xiong Zhang** [1,3], **Jing Li** [1,2], **Yixue Jiao** [1,3], **Haowen Feng** [1,2] and **Yuhua Wang** [1,*]

[1] Yunnan Key Laboratory for Wild Plant Resources, Kunming Institute of Botany, Chinese Academy of Sciences, Kunming 650201, China; dingxiaoyong@mail.kib.ac.cn (X.D.); guochangan@mail.kib.ac.cn (C.G.); zhangxiong@mail.kib.ac.cn (X.Z.); lijing@mail.kib.ac.cn (J.L.); jiaoyixue@mail.kib.ac.cn (Y.J.); fenghaowen@mail.kib.ac.cn (H.F.)

[2] University of Chinese Academy of Sciences, Beijing 100049, China

[3] College of Life Sciences, Shaanxi Normal University, Xi'an 710119, China

[*] Correspondence: wangyuhua@mail.kib.ac.cn

[†] These authors contributed equally to this work.

**Abstract:** This study documented the wild plants used by Tibetans and the related traditional knowledge in Burang Town (Karnali River Valley). Ethnobotanical surveys, including semi-structured interviews and participatory observations, were conducted in five Tibetan communities in July 2020 and August 2021. The informant consensus factor (ICF) and cultural importance index (CI) were used for data analyses. In total, 76 wild species belonging to 58 genera and 30 families were determined to be used. These included 26 edible, 29 medicinal, 34 fodder, 21 fuel, 17 incense, three economic, three dye, two ritual, two handicraft, and one species for tobacco plants species; many of these have multiple uses. The top five important plants are *Carum carvi* (CI = 1.88), *Hippophae tibetana* (CI = 1.45), *Rheum moorcroftianum* (CI = 0.87), *Urtica dioica* (CI = 1.45) and *Chenopodium album* (CI = 0.75). Of the wild plant species used, 53 were recorded in croplands and 25 were found in alpine pastures. This pattern of use is influenced by local livelihood patterns and culture. Plants in high-land cropland have diverse ethnobotanical values that are often overlooked. These findings will inform strategies and plans for local communities and governments to sustainably use and protect plants at high altitudes.

**Keywords:** wild plants; traditional knowledge; Karnali River Valley; cropland; Tibetan; Burang Town

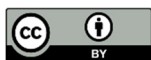

## 1. Introduction

Various wild plants are an important part of the culture and livelihood of the people in the Himalayas, as they are used by the local people for food, medicine, fodder, fuel, dyes, rituals, and other products and services [1–7]. As poor biophysical conditions, inconvenient transportation, limited arable land, and seasonal economy are significant challenges to food and nutrition security in the Himalayas [8], wild edible plants are an important source of carbohydrates and nutrition for local people, especially in such high-altitude areas [1,9–11]. Healthcare of the local communities is also an important challenge and traditional medicine has always been a means of health care in the local community [12–15]. Various Himalayan plants act as free pharmacies for locals. Notably, as wild plants are also an economic resource, their large-scale collection also has a certain impact on the population of these plants [16–18]. However, an important problem is that a large area of the Himalayas is not covered by forests, and that there are large areas of alpine meadows, such as the Ngari area in China [19]. Compared with the communities living

in or close to forests, few ethnobotanical studies have examined the non-forest areas in high mountains [2,3].

Tibetans, the main ethnic group in the Qinghai-Tibet Plateau (QTP), are one of the 56 ethnic minorities in China. Historically, Tibet has been divided into three cultural regions based on language (Kham, Ü-Tsang, and Amdo Tibet) [20]. However, ethnobotanical research on Tibetans has been limited compared to that on the surrounding biological and cultural diversity areas because of factors such as geographical isolation and inconvenient transportation [21]. The current ethnobotanical research on Tibetans is mainly concentrated in the Hengduan Mountains and the eastern part of the QTP in China [2,9,21–24]. The number of plants and species used by Tibetans living in different environments varies greatly [2,9,21,22], possibly because of the differences in vegetation in different geographic regions [25]. However, the structures of edible plant vegetables and fruits at different survey sites are similar [9,21]. Some species that grow in high-altitude areas or are widely distributed (such as *Urtica* sp., *Chenopodium albumare* and *Thymus* sp.) are selected and used by Tibetans [2,9,21,22]. Compared with these research areas, it is unclear whether similar plants are used by Tibetans in high-altitude areas of western China.

The Karnali River Valley, a Tibetan settlement, is located in Burang County, Tibet, China. It is geographically located at the junction of the central and western Himalayas and is one of the 23 vertical fault valleys in the Himalayas [26]. The average elevation of both banks of the river valley is approximately 3900 m, which typical of a high-altitude area and the natural vegetation is an alpine desert meadow [18]. This region has a long history and culture. It has the world-renowned mountain, Kangrinboqe, which is the sacred to Yongzhong Bonism, Hinduism, Jainism, and Tibetan Buddhism. It is also one of the core areas of the ancient Xiangxiong Kingdom. Moreover, these region has been an important channel for exchange and trade between Tibet and the South Asian subcontinent since ancient times [26]. Burang Tibetans speak the Ü-Tsang dialect, but their dialect has strong specific characteristics [27].

Considering the lack of detailed information regarding the plants used in this region, this study aimed to document the knowledge of traditional usage of plants by Tibetans in Karnali River Valley and to explore the characteristics of the use of plants by Tibetans living at different places. We aimed to answer the following questions: (1) Which plants are used by Tibetans in the Karnali River Valley? (2) What are the characteristics of the plant used? We hypothesised that the number of edible plants in the Karnali River Valley is less than that in the southern QTP, but the structure of the plants used is similar, and there are few plants that are used commonly in different regions.

## 2. Materials and Methods

### 2.1. Study Sites

Ngari Prefecture is the only regional administrative region in the Tibet Autonomous Region. It is located in the northern part of the Qinghai-Tibet Plateau (Qiangtang Plateau) with an average altitude of more than 4500 m. It is called the "Forbidden Zone of Life". Burang Town belongs to Burang County, Ngari Prefecture, Tibet Autonomous Region. It is located in the southernmost part of Ngari Prefecture and borders India and Nepal. It belongs to the temperate arid climate zone of Ali Plateau, with an altitude of 3800–4500 m. In these region, there are 50–100 days when the average daily temperature is 10 °C; the average temperature of the warmest month is 10–12 °C; further, the winter is severely cold with little snow, and the annual precipitation is 50–100 mm [27]. Vegetation in the Karnali River Valley is diverse. The region from 3600–4100 m is a desert grassland with *Caragana gerardiana* shrubs and *Stipa ceratoides* (L.) Gueldenst. *Thymus* grasslands are also found on the local slopes at approximately 4000 m. The region from 4100–4500 m is a desert steppe of *S. ceratoides*, with patches of *C. versicolor* Benth. shrubs in the valleys and foothills. The region from 4600–5000 m is a shrub grassland composing of *C. versicolor* Benth. and *S. purpurea*; 4800–5000 m is distributed with patches of *Rhododendron anthopogon* D. Don

shrubs in shaded and humid places. The part from 5000–5500 m has a *Kobresia myosuroides* alpine meadow. Above 5500 m are snow-covered screes [19].

The Karnali River (孔雀河) originates from glaciers in the northern part of Guzhenla Mountain in Burang County of the Himalayas in Tibet. It is one of the "Four Sacred Rivers" of Ngari along with the Shiquan River (狮泉河), Xiangquan River (象泉河), and Maquan River (马泉河). The Wharf port flows into Simikot County, Nepal, and finally into the Ganges River [27]. In China, under the influence of the humid ocean monsoon in the Indian Ocean, the Karnali River Valley has a unique climate with warmth and abundant rainfall, which makes Burang Town a rare area suitable for cropping in Ngari. The terrain on both sides of the basin is low, approximately 3900 m above sea level (Figure 1). We investigated all five administrative villages in Burang Town: Kejia Village, Xide Village, Chide Village, Duoyou Village, and Rengong Village. These villages are scattered on both sides of Karnali River. The resident population of Burang Town is almost entirely Tibetan, with a total population is 7777 based on the 2020 demographic data. The economy in this area is underdeveloped, mainly based on animal husbandry and agriculture. The main livestock type in grazing systems is yak [28]. The planting system mainly revolves around crops, such as highland barley, potatoes, peas, and early maturing rape, which ripened once a year. Figure 2 presents the landscape of Burang Town at different altitudes (cropland and alpine pasture).

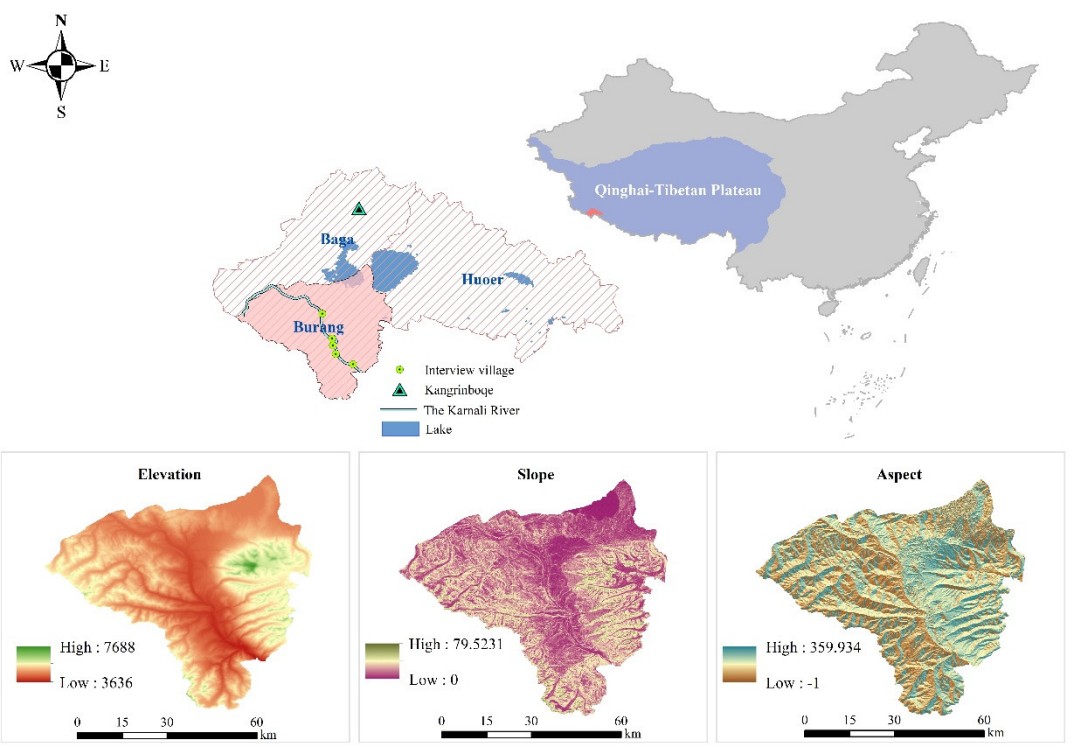

**Figure 1.** The study area: Burang Town, Tibet, China.

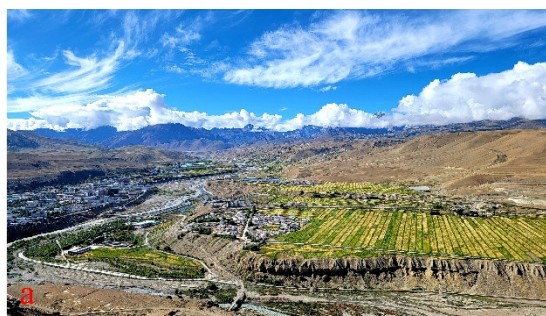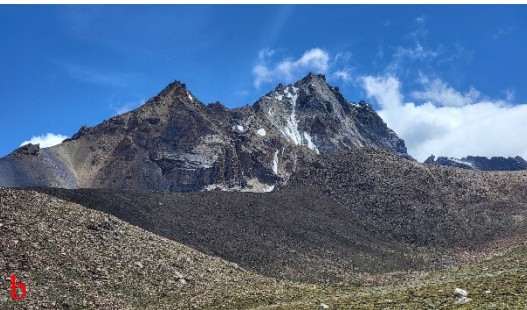

**Figure 2.** The landscape of Burang Town at different altitudes. (**a**): cropland; (**b**): alpine pasture.

*2.2. Ethnobotanical Field Study*

The field studies were conducted within the five Tibetan communities of Burang Township in July 2020 and August 2021 (Figure 1). In total, 99 informants were interviewed (Table 1). First, we visited the township committee to obtain permission for the fieldwork. We explained our purpose to committee leaders and asked for help, which mainly included obtaining local guides, introducing us to local villagers and other necessary assistance. Wild plants and related local knowledge were collected through semi-structured interviews and direct observations. All field works were carried out with informed consent. Considering the relative lack of educational resources, most local people, especially aged community members, could not communicate fluently in mainstream Chinese. Therefore, field works were performed with the assistance of local guides. The informants of semi-structured interviews were selected randomly from local community members, and older people were deliberately chosen as priority informants. Semi-structured interviews were used to obtain information regarding the local knowledge of wild plants, which were provided by local people. The vernacular names, life forms, parts used, uses, collection seasons, economic values, and cultural significance of the plants were documented. Images and video data of the local uses of wild plants were documented with a digital camera for direct observation. Voucher specimens were also collected. The semi-structured interviews were conducted based on the following questions: (1) What wild plants do you use in daily life for food, medicine, fodder, incense, or other purposes? (2) What parts do you use? How do you prepare these? (3) Where do you collect these plants, and when are they collected? For each plant, we also asked the informants to list the specific collection area (crop land or alpine pastures) for the plant.

During the field work, with the help of local guides, we collected plant specimens considering that one plant corresponds to one vernacular name. Photographs of the plants were also taken to identify the scientific taxa of plants later. Specimens were then identified and stored in the herbarium of the Kunming Institute of Botany, Chinese Academy of Sciences (KUN). Plant species were identified according to the Flora of Xizang and Flora of China [29,30]. The Latin names of the plants were proofread based on the plant list [31].

**Table 1.** Characteristics of informants.

| Characteristics | Number | Number |
| --- | --- | --- |
| Villages | Rengong Village | 21 |
| | Duoyou Village | 22 |
| | Chide Village | 11 |
| | Xide Village | 28 |
| | Kejia Village | 17 |
| Gender | Female | 40 |
| | Male | 59 |
| Age groups | 21–40 | 8 |
| | 41–60 | 47 |

| | | |
|---|---|---|
| | Above 60 | 45 |
| Education level | Illiterate | 31 |
| | Primary | 59 |
| | Secondary | 6 |
| | High school | 1 |
| | above | 2 |

*2.3. Data Analysis*

The informant consensus factor (*ICF*) and cultural importance index (*CI*) were used as ethnobotanical quantitative indices. The *ICF* was used to evaluate the consensus of Tibetans in the treatment of specific diseases [32].

$$ICF = \frac{Nur - Nt}{Nur - 1}$$

where *Nur* is the number of use reports mentioned by the informants to treat a certain type of disease. *Nt* is the number of plants used by the informant to treat a specific disease. The value of the *ICF* ranges from 0–1. The higher the value, the higher is the consensus on the hypothesis of using a given treatment method to treat a given disease. Moreover, the *ICF* can also provide a "remedy of a choice" that evaluates whether a certain plant is more likely to be an effective treatment for a given disease.

To evaluate the comprehensive utilisation value of each wild plant used by the Tibetans living in the Karnali River Valley, we calculated the *CI* value. The cultural importance index (*CI*) was defined as the sum of the percentage of respondents who mentioned various uses for a certain useful plant [33]. Additionally, the *CI* considers the various uses of each plant, and the dissemination of knowledge (for each use-category of each plant). In other words, the diversity of the plant uses and the degree of recognition among the informants for each use-category were included. This can indicate the comprehensive utilisation value of each useful plant. The *CI* was calculated using the following formula:

$$CI = \sum_{u=u_1}^{u_{NC}} \sum_{i=i_1}^{i_N} \frac{UR_{ui}}{N}$$

where *N* is the number of use reports, and *NC* is the number of use categories. *i* and *u* represent the informant and the use category of the plant, respectively. A high *CI* value indicates that a certain plant has multiple uses and is widely known.

## 3. Results

*3.1. Diversity of Wild Useful Plants and Their Habitats*

We documented a total of 76 wild species belonging to 58 genera and 30 families, which were used by Tibetans in Karnali River Valley (Table 2). The results showed that the most frequently mentioned families were Compositae (17 species), followed by Rosaceae (six species). The families Leguminosae, Polygonaceae and Ranunculaceae each had four species whereas Salicaceae, Caprifoliaceae, and Amaranthaceae each had three species. Ten families had two species each. The remaining 12 families had only species each. At the genus level, the most common genus was *Artemisia* (seven species), followed by *Potentilla* (five species) and *Salix* (three species). Of these plant species, *Neopicrorhiza scrophulariiflora* and *Juniperus tibetica* are listed in the IUCN Red List [34]. Of the taxa, 59 were herbaceous plants (77.6%), 14 were shrubs (18.4%) and three were trees (4.0%). Among these plants, *Allium przewalskianum* has been introduced in home gardens by the local people.

Agriculture and animal husbandry are the main livelihoods of the Tibetans in Burang Town, and the collection areas (mainly in cropland and alpine pastures) of wild plants use are also closely related to their livelihoods. Figure 3 shows the number of plants collected

from the cropland and alpine pasture. Among them, 53 wild plants were collected and used in croplands, accounting for 69.7% of all useful plants. The number of wild plants used in croplands is double that in alpine pastures.

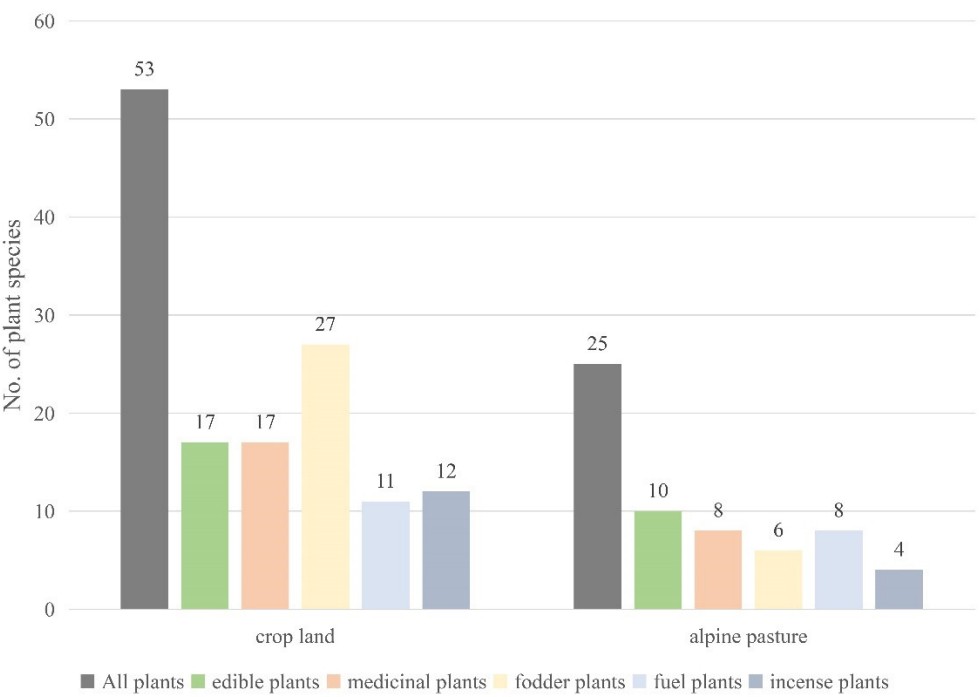

**Figure 3.** Collection sites of all used plants, including edible, medicinal, fodder, fuel, and incense plants in the study area.

**Table 2.** Wild plants used by Tibetans in Burang Town.

| Family | Species | Venecular Name (Tibetan Script) | Local Use (Parts Used) | Collection Site | CI | Voucher Number |
|---|---|---|---|---|---|---|
| Amaranthaceae | *Chenopodium album* L. | niu (ནུཝ) | Food: cooked vegetable (stems and leaves). Medicinal: consumed with yogurt to treat motion sickness or as restorative (stems and leaves). Fodder: yak feed (aerial parts). | cropland | 0.75 | EBT-PL-24 |
| Amaranthaceae | *Salsola tragus* L. | ye si (ཡད་སི) | Food: cooked vegetable (tender stems and leaves). Fodder: yak feed (aerial parts). | cropland | 0.12 | EBT-PL-2 |
| Amaranthaceae | *Krascheninnikovia ceratoides* (L.) Gueldenst. | jia bu xin (ཅ་བུའི་ཞིན) | Fodder: yak feed (aerial parts). Fuel: used to strike a fire (aerial parts). | cropland | 0.07 | EBT-PL-34 |
| Amaryllidaceae | *Allium przewalskianum* Regel | zen bu (བཟེན་སྦུག) | Food: seasoning, cooked vegetable (whole plants). | alpine pasture | 0.74 | EBT-PL-50 |
| Amaryllidaceae | *Allium carolinianum* DC. | ri guo (རི་གོད), guo-ba-ri-guo (གོད་པ་རི་གོད) | Food: seasoning, cooked vegetable (whole plants). | cropland | 0.39 | EBT-PL-3 |
| Apiaceae | *Carum carvi* L. | kuo nie (ཀྲུའ་ནེ), kuo-li-ma (ཀྲུའ་ལི་མ) | Food: cooked vegetable (tender aerial parts); seasoning (seeds). Medicinal: used to treat headache, toothache, hypertension, mountain | cropland | 1.88 | EBT-PL-37 |

| Family | Species | Local name | Uses | Habitat | | Voucher |
|---|---|---|---|---|---|---|
| | | | sickness (seeds); cooked and mixed with tsampa to treat arthritis (tender aerial parts). | | | |
| Apiaceae | *Vicatia thibetica* H. Boissieu | pang bu (པང་སྦུག) | Fodder: yak feed (aerial parts). | cropland | 0.05 | EBT-PL-11 |
| Boraginaceae | *Arnebia euchroma* (Royle) Johnst. | mu zi (སྨུག་པཟེ) | Medicinal: liniment, used to treat nappy rash, baldness, acne (roots). Dye plant (roots). Ritual plant (roots). Incense plant (roots). | cropland | 0.29 | EBT-PL-52 |
| Brassicaceae | *Christolea crassifolia* Cambess. | tu zi ra (སྨུག་བཟེ་ར) | Fodder: yak feed (aerial parts). Fuel: used to strike a fire (aerial parts). | cropland | 0.03 | EBT-PL-64 |
| Caprifoliaceae | *Lonicera spinosa* (Decne.) Jacq. ex Walp. | zhai (གྲད), jie gong zhong (ཅད་གོན་ཐོན) | Food: fruit (fruits). Fuel: fuelwood (whole plants). | alpine pasture and cropland | 0.41 | EBT-PL-42 |
| Caprifoliaceae | *Nardostachys jatamansi* (D. Don) DC. | bang bu (པང་སྦུག) | Incense plant (rhizomes). | alpine pasture | 0.38 | EBT-PL-100 |
| Caprifoliaceae | *Lonicera rupicola* Hook. f. & Thomson | zhai (གྲད) | Food: fruit (fruits). Fuel: fuelwood (whole plants). | alpine pasture and cropland | 0.23 | EBT-PL-41 |
| Caryophyllaceae | *Lepyrodiclis holosteoides* (C.A. Mey.) Fenzl ex Fisch. & C.A. Mey. | men duo guo ra (མེན་ཏུའོ་གོ་ར) | Food: cooked vegetable (tender aerial parts). Medicinal: liniment, emollient (leaves). Fodder: yak feed (whole plants). | cropland | 0.42 | EBT-PL-53 |
| Caryophyllaceae | *Silene moorcroftiana* Wall. ex Benth. | che ru (ཀྲུ་ཟུག) | Fuel: kindling (whole plants). | alpine pasture | 0.04 | EBT-PL-32 |
| Compositae | *Waldheimia glabra* (Decne.) Regel | kang ba (ཁང་པ) | Medicinal: liniment, the flowers are boiled in water or the leaves are crushed and applied on the face to treat infestations and poisonings due to: bites and stings (leaves and flowers). Incense plant (whole plants). Fuel: kindling (whole plants). | cropland | 0.69 | EBT-PL-62 |
| Compositae | *Artemisia roxburghiana* Bess. | bu r kan ba (སྦུ་ར་ཁན་པ) | Medicinal: smoke therapy to treat lnfections (aerial parts). Incense plant (aerial parts). Ritual plant: funeral ceremony (aerial parts). Fuel plant (aerial parts). Fodder (tender aerial parts). | cropland | 0.42 | EBT-PL-60 |
| Compositae | *Artemisia desertorum* Spreng. | pu lu mo (སྤུ་ལུའུ་མོ), chuo bu (ཆུའོ་སྦུག) | Medicinal: smoke therapy to treat lnfections (aerial parts). Incense plant (aerial parts). | cropland | 0.36 | EBT-PL-99 |
| Compositae | *Saussurea tridactyla* Sch.Bip. ex Hook.f. | gang la mei duo (གངས་ལ་མེ་ཏོག) | Medicinal: medical liquors, liniment, used to treat arthritis (aerial parts). Economic plant (aerial parts). | cropland | 0.31 | EBT-PL-21 |

| Compositae | *Aster acer* Roehl. | kang ga bu (ཁང་ག་ ཕུབ།) | Medicinal: smoke therapy to treat lnfections (aerial parts). Incense plant (aerial parts). | alpine pasture | 0.28 EBT-PL-18 |
|---|---|---|---|---|---|
| Compositae | *Artemisia* sp. | kan ba la bu (ཀན་པ་ ལ་ཕུབ།) | Incense plant (aerial parts). | cropland | 0.12 EBT-PL-27 |
| Compositae | *Artemisia demissa* Krasch. | kan ba sei bu (ཀན་ པ་སྱུམ་ཕུབ།) | Incense plant (aerial parts). | cropland | 0.12 EBT-PL-7 |
| Compositae | *Artemisia annua* L. | kan ba la ge lang (ཁན་པ་ལ་གི་ལང་།) | Incense plant (aerial parts). Fodder (tender aerial parts). | cropland | 0.11 EBT-PL-8 |
| Compositae | *Saussurea medusa* Maxim. | gang la mei duo (གངས་ལ་མེ་ཏོག) | Medicinal: medical liquors, liniment, used to treat arthritis (aerial parts). Economic plant (aerial parts). | alpine pasture | 0.09 EBT-PL-12 |
| Compositae | *Cirsium arvense* (L.) Scop. | da zi la (ཏ་ཟི་ལ།), ci ma (ཚི་མ།) | Fodder: yak feed (aerial parts). | cropland | 0.05 EBT-PL-9 |
| Compositae | *Hypochaeris ciliata* (Thunb.) Makino | mo nu (མོ་ནུབ།) | Incense plant (aerial parts). | cropland | 0.04 EBT-PL-6 |
| Compositae | *Carlina biebersteinii* Bernh. ex Hornem. | jiang zi (ཅང་ཟི།) | Fodder: yak feed (aerial parts). | cropland | 0.04 EBT-PL-25 |
| Compositae | *Leontopodium leontopodioides* (Willd.) Beauv. | zha (ཞ།) | Fuel: kindling (whole plants). | alpine pasture | 0.04 EBT-PL-23 |
| Compositae | *Taraxacum eriopodum* (D.Don) DC. | dia dong (ད་དུང་།) | Medicinal: potions, used to treat gastritis (whole plants). | cropland | 0.04 EBT-PL-83 |
| Compositae | *Artemisia stracheyi* Hook. f. et Thoms. | kan jiong ga bu (མཁན་པ་ཇ་པོ།) | Incense plant (aerial parts). | cropland | 0.02 EBT-PL-5 |
| Compositae | *Artemisia gmelinii* Web. ex Stechm. | kan ba sei bo (མཁན་ པ་སེའི་པོ།) | Incense plant (aerial parts). | cropland | 0.02 EBT-PL-102 |
| Compositae | *Melanoseris macrorhiza* (Royle) N. Kilian | wang za (ཝང་བཟའ།) | Fodder: yak feed (aerial parts). | cropland | 0.02 EBT-PL-63 |
| Convolvulaceae | *Convolvulus arvensis* L. | ga la you wa (ག་ལ་ ཡོའུ་བ།) | Food: cooked vegetable (tender stems and leaves). Fodder: yak feed (aerial parts). | cropland | 0.33 EBT-PL-57 |
| Cupressaceae | *Juniperus indica* Bertoloni | xiu bai (ཤུག་པའི།) | Medicinal: smoke therapy to treat lnfections (aerial parts). Incense plant (aerial parts). Fuel: firewood (whole plants). | cropland | 0.47 EBT-PL-15 |
| Cupressaceae | *Juniperus tibetica* Kom. | che xiu (ཚེ་ཤུག) | Incense plant (aerial parts). | cropland | 0.03 EBT-PL-16 |
| Elaeagnaceae | *Hippophae tibetana* Schltdl. | da ge jia ma (ཏ་གི་ཅ་ མ།), dai ru wa (ཏའི་ རུའ་བ།) | Food: fruit (fruits); alternatives to Vinegar (fruits). Medicinal: liniment, the fruit is boiled and smeared to treat gastritis and mountain sickness (fruits). Fuel: firewood (stems). Fodder: yak feed (leaves). | cropland | 1.45 EBT-PL-51 |
| Ephedraceae | *Ephedra gerardiana* Wall. ex Stapf | ci (ཚི།) | Food: fruit (fruits). Medicinal: pulverised/powdered, calcinated, allergic rhinitis. | alpine pasture | 0.07 EBT-PL-45 |
| Ephedraceae | *Ephedra saxatilis* Royle ex Florin | ci (ཚི།) | Food: fruit (fruits). Medicinal: pulverised/powdered, calcinated, | cropland | 0.07 EBT-PL-17 |

| | | | allergic rhinitis. Fuel: firewood and kindling (aerial parts) | | | |
|---|---|---|---|---|---|---|
| Equisetaceae | *Equisetum arvense* L. | di ge zu ba (དུས་གི་ ཚུའ་བ།), ke ma (ཁུ་མ།) | Fodder: yak feed (aerial parts). | cropland | 0.12 | EBT-PL-19 |
| Ericaceae | *Rhododendron anthopogon* D. Don | po lu (པོ་ལུ།) | Medicinal: smoke therapy to treat lnfections (aerial parts). Incense plant (aerial parts). Fuel: firewood (whole plants). | alpine pasture | 0.56 | EBT-PL-91 |
| Ericaceae | *Rhododendron lepidotum* Wall. ex G. Don | se lu (སེ་ལུ།) | Incense plant (aerial parts). | alpine pasture | 0.2 | EBT-PL-92 |
| Gentianaceae | *Swertia ciliata* (D. Don ex G. Don) B.L. Burtt | di g da (ཏིས་གི་ཏ), gei di (གེ་ཏི།) | Medicinal: potions, used to treat headache and biliiousness (whole plants). Fodder: yak feed (whole plants). | cropland | 0.25 | EBT-PL-104 |
| Gentianaceae | *Gentiana veitchiorum* Hemsl. | bang jian mei duo (པང་ཅན་མེ་ཏོག) | Medicinal: potions, used to treat lnfections (whole plants). | alpine pasture | 0.01 | EBT-PL-20 |
| Juncaceae | *Juncus thomsonii* Buchenau | bo ru (པོ་རུག) | Food: fruit (rhizomes). | cropland | 0.04 | EBT-PL-103 |
| Lamiaceae | *Thymus linearis* Benth. | guo ma ra za (གོ་མ་ར་ཙ), gang ma ra (གང་མ་ར།) | Food: seasoning (whole plants). Medicinal: potions, used to treat headache and biliiousness (whole plants). Fodder: yak feed (leaves). Economic plant (whole plants). | alpine pasture | 0.67 | EBT-PL-48 |
| Lamiaceae | *Dracocephalum heterophyllum* Benth. | jiang gu gu ru (ཅང་ གུའ་གུའ་རུག) | Food: fruit (flowers). Fodder: yak feed (aerial parts). | cropland | 0.13 | EBT-PL-4 |
| Leguminosae | *Cicer microphyllum* Benth. | cei jiu wa (སུས་ཅུ་བ།) | Food: fruit (fruits); cooked vegetable (fruits). Fodder: yak feed (leaves). | cropland | 0.6 | EBT-PL-88 |
| Leguminosae | *Caragana versicolor* Benth. | chang (ཆང་།) | Fodder: yak feed (leaves). Fuel: firewood (whole plants). | alpine pasture | 0.54 | EBT-PL-93 |
| Leguminosae | *Oxytropis tragacanthoides* DC. | na da la (ན་ཏ་ལ།) | Food: fruit (fruits); cooked vegetable (fruits). Incense plant (aerial parts). Fuel: kindling (aerial parts). | alpine pasture | 0.48 | EBT-PL-13 |
| Leguminosae | *Astragalus polycladus* Bureau & Franch. | da ge xia (ཏ་གི་ཞ།) | Medicinal: liniment (whole plants). | cropland | 0.02 | EBT-PL-70 |
| Leguminosae | *Melilotus officinalis* (L.) Pall. | niu ga liu ba (ནུཕ་ག་ ལུཕ་བ།) | Fodder: yak feed (leaves). | cropland | 0.01 | EBT-PL-71 |
| Malvaceae | *Malva verticillata* L. | chi ma han di (ཁུ་མ་ ཧན་ཏི།) | Fodder: yak feed (leaves). | cropland | 0.01 | EBT-PL-78 |
| Orchidaceae | *Gymnadenia conopsea* (L.) R.Br. | sei xia (སེ་ཞ།), ang bu la ba (འང་ཕུ་ལ་བ།) | Medicinal: potions, restorative (roots). | alpine pasture | 0.15 | EBT-PL-105 |
| Plantaginaceae | *Plantago depressa* Willd. | ta rang (ཐ་རང་།), ga la (ག་ལ།) | Medicinal: potions, infections and haemorrhages of pregnancy (whole plants). Fodder: yak feed (whole plants) | cropland | 0.04 | EBT-PL-106 |
| Plantaginaceae | *Neopicrorhiza scrophulariiflora* (Pennell) D.Y.Hong | hong lei (ཧོན་ལེའི།) | Medicinal: potions, infections (roots). | alpine pasture | 0.03 | EBT-PL-54 |

| Poaceae | *Polypogon fugax* Nees ex Steud. | rang ba (རང་བ།) | Fodder: yak feed (leaves). | cropland | 0.37 | EBT-PL-31 |
|---|---|---|---|---|---|---|
| Poaceae | *Pennisetum flaccidum* Griseb. | ruan ba (རན་བ།) | Fodder: yak feed (leaves). | cropland | 0.36 | EBT-PL-67 |
| Polygonaceae | *Rheum moorcroftianum* Royle | de jiu wa (ཏེ་རྒྱུའ་བ།) | Food: fruit, to be eaten directly after peeling (stems). Medicinal: liniment, arthritis (roots). | alpine pasture | 0.87 | EBT-PL-35 |
| Polygonaceae | *Polygonum affine* D. Don | meng zhu wa (མེན་གྱུའ་བ།) | Food: staple food, seed powder used to make tsampa (seeds). Fodder: yak feed (whole plants). | alpine pasture | 0.31 | EBT-PL-33 |
| Polygonaceae | *Rumex nepalensis* Spreng. | ma ra ma (མ་ར་མ།) | Food: cooked vegetable (leaves). Medicinal: liniment, demulcent (roots). Fodder: yak feed (whole plants). | cropland | 0.27 | EBT-PL-36 |
| Polygonaceae | *Polygonum tortuosum* D. Don | nia luo (ན་ལༀ།) | Dye plant (aerial parts). Other: tobacco substitute (leaves). | alpine pasture | 0.23 | EBT-PL-86 |
| Ranunculaceae | *Clematis tangutica* (Maxim.) Korsh. | bian ma (པན་མ།) | Fodder: yak feed (whole plants). | cropland | 0.28 | EBT-PL-56 |
| Ranunculaceae | *Delphinium kamaonense* Huth | peng ma (ཕེན་མ།) | Meddicinal: liniment, rashes/heat rash, intoxication (tubers). Fodder: yak feed (whole plants). | alpine pasture and cropland | 0.1 | EBT-PL-40 |
| Ranunculaceae | *Aconitum pendulum* N.Busch | peng ma (ཕེན་མ།) | Fodder: yak feed (whole plants). | cropland | 0.07 | EBT-PL-30 |
| Ranunculaceae | *Delphinium nordhagenii* Wendelbo | bo za (པོ་བཟའ།) | Fodder: yak feed (whole plants). | alpine pasture | 0.05 | EBT-PL-49 |
| Rosaceae | *Potentilla anserina* L. | chu wa (གྱུའ་བ།) | Food: staple food, a substitute for tsampa (tubers). | cropland | 0.48 | EBT-PL-44 |
| Rosaceae | *Potentilla parvifolia* Fisch. ex Lehm. | bian ma (པེན་མ།) | Fuel: firewood (whole plants). Others: used to make agricultural tools (branches). | alpine pasture | 0.41 | EBT-PL-43 |
| Rosaceae | *Rosa sericea* Wall. ex Lindl. | sei you la (སྲུས་ཡུའ་ལ།) | Food: fruit (fruits). Fuel: firewood (whole plants). | cropland | 0.23 | EBT-PL-97 |
| Rosaceae | *Potentilla fruticosa* var. *pumila* Hook.f. | bian ma (པེན་མ།) | Food: beverage, tea substitute, and leaves soaked in water (leaves). Others: used to make agricultural tools (branches). Fuel: firewood (whole plants). | alpine pasture | 0.18 | EBT-PL-22 |
| Rosaceae | *Potentilla bifurca* L. | qia chuo luo (ཆ་ཁྲོ་ལྀ།) | Food: cooked vegetable (leaves). Fodder: yak feed (whole plants). | cropland | 0.08 | EBT-PL-14 |
| Rosaceae | *Potentilla saundersiana* Royle | qiu gu mei duo (ཆྱུ་གུའ་མེ་ཏྀག) | Food: beverage, tea substitute, and leaves soaked in water (leaves and flowers). Fodder: yak feed (whole plants). | alpine pasture | 0.05 | EBT-PL-65 |
| Salicaceae | *Salix matsudana* Koidz. | jiang ma (ཅང་མ།) | Food: cooked vegetable (tender shoot). Fuel: firewood (whole plants). | cropland | 0.32 | EBT-PL-38 |
| Salicaceae | *Salix sclerophylla* Andersson | lang ma (ལིང་མ།) | Fuel: firewood (whole plants). | cropland | 0.16 | EBT-PL-26 |

| Salicaceae | *Salix sericocarpa* Andersson | lang ma (ཤིང་མ།) | Fuel: firewood (whole plants). | cropland | 0.14 | EBT-PL-107 |
|---|---|---|---|---|---|---|
| Solanaceae | *Hyoscyamus niger* L. | lang dang (ལང་ཏིང་།) | Medicinal: smoke therapy, burned, toothache (seeds). Fodder: yak feed (whole plants). | cropland | 0.18 | EBT-PL-75 |
| Tamaricaceae | *Myricaria prostrata* Hook. f. & Thomson | ong bu (ཨོན་པུ།) | Medicinal: veterinary, enemas, pneumonia, intoxication ((aerial parts). Fuel: firewood (whole plants). | cropland | 0.27 | EBT-PL-72 |
| Urticaceae | *Urtica dioica* L. | sa (ས།), sa bu (ས་པུ།) | Food: cooked vegetable, stuffing for buns (tender leaves). Medicinal: medicinal foods, restorative, used to treat diarrhoea (tender leaves). Fodder: yak feed (whole plants). | cropland | 0.78 | EBT-PL-100 |

*3.2. Use Categories of Wild Useful Species*

Of 76 wild plant species in total, many plants have multiple uses, and 57.9% (44 species) of these plants belonging to two to five use categories. These plants are used to meet the daily product and service needs of by the Burang Tibetans. Our results showed that there were 26 edible plant species, 29 medicinal plant species, 34 fodder plant species, 21 fuel plant species, 17 incense plant species, three economic plant species, three dye plant species, two ritual plant species, two handicraft plant species, and one tobacco plant species (Figure 4).

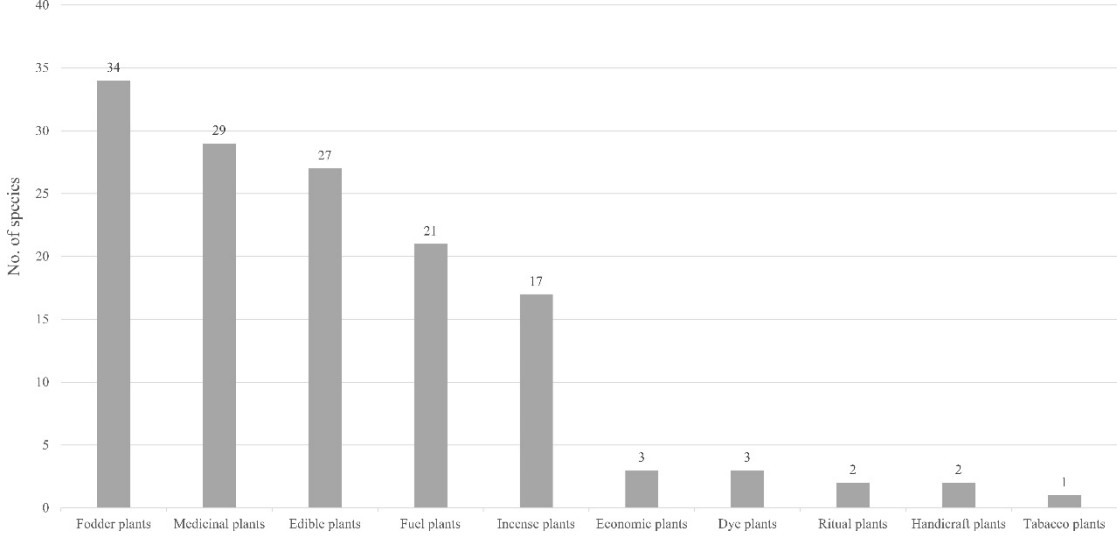

**Figure 4.** Plants used for different purposes in the study area. The abscissa indicates different purposes, and the ordinate indicates the number of plant species corresponding to different purposes.

3.2.1. Wild Edible Plants

In this study, 27 wild plant species were identified as edible. Based on the dietary habits of Tibetan people, these plants are used as vegetables, fruits, seasoning, staple foods, and beverages. Among these categories, most species were vegetables (14 species), followed by fruits (11 species), seasoning (five species), stable foods (two species), and beverages (two species). The wild edible plants most frequently used as vegetables by the Tibetan people play a vital role in supplementing minerals, vitamins, and fiber to the inhabitants of Karnali River Valley. For example, *Carum carvi*, *Chenopodium album*, and *Urtica dioica* are important vegetables for local Tibetan people, and are generally cooked as

fried dishes or soups after blanching in boiling water and draining. The young leaves of these plants are usually collected and dried for consumption in winter. Wild fruits are another important food source in the daily life of Burang Tibetans. Wild fruits, such as *Hippophae tibetana*, *Lonicera spinosa*, *Cicer microphyllum*, and *Rosa sericea* are commonly consumed directly. Further, some informants mentioned the rhizomes of *Juncus thomsonii* as a fruit. This is the first report of this plant as an edible plant. In general, vegetables and fruits are the most abundant edible plants and are widely used in Karnali River Valley. This result is consistent with the results of several previous ethnobotanical surveys on Tibetans, which reported that wild vegetables and fruits constitute an important source of nutrition for Tibetans living in different regions [2,9,21,22]. Wild plants are also commonly used as seasoning plants. For example, the leaves of *Allium przewalskianum* and *Thymus linearis* are most commonly sprinkled on vegetables or soups to increase the flavour (Figure 5a–c). Moreover, the seeds of *Polygonum affine* and tubers of *Potentilla anserina* are used as staple foods (Figure 5d); further, the flowers of *Potentilla saundersiana* and *Potentilla fruticosa* var. *pumila* are boiled in water to prepare tea.

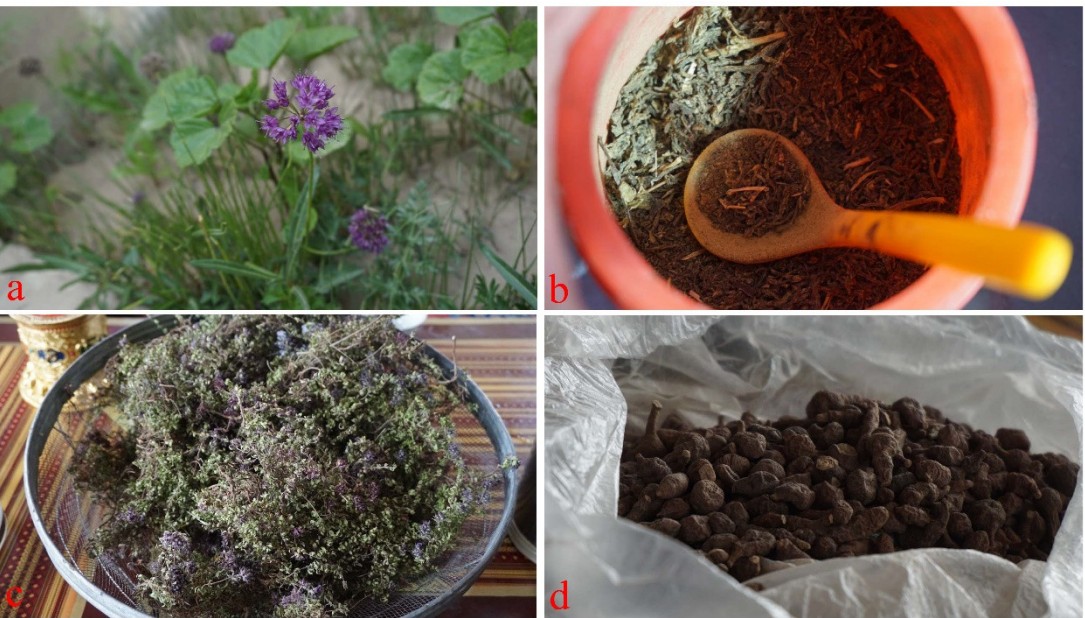

**Figure 5.** Three wild edible plants. (**a**,**b**): *Allium przewalskianum*; (**c**): *Thymus linearis*; (**d**): *Potentilla anserina*.

### 3.2.2. Wild Medicinal Plants

During the investigation, we documented 29 medicinal plants used by the Tibetan people in Karnali River Village. The greatest proportion of medicinal plants included herbs (79.3%), followed by shrubs (20.7%). The parts of the medicinal plants used included overground parts (34.5%), followed by whole plants (20.7%), roots (20.7%), leaves (13.8%), tender leaves (10.3%), flowers (3.4%), seeds (3.4%), and fruits (3.4%). There are six types of prepared traditional medicines: liniments, potions, smoke therapy, medicinal foods, pulverised parts/powders, and enemas. Liniment is the most commonly used type of traditional medicine prepared from wild plants (10 plant species), followed by potions (eight species). Five plants were used for smoke therapy. Three species were prepared as medicinal foods. Two species were pulverized/powdered. Only one plant was used as enemas. In contrast to most studies in which decoction was the main preparation method, liniment was the main preparation method of medicinal plants in Burang [35].

According to the information provided by the informants and the Economic Botany Data Collection Standard, diseases treated using medicinal plants are divided into 10 categories [36]. These included pregnancy/birth/puerperium disorders, circulatory system

disorders, pain, muscular-skeletal system disorders, digestive system disorders, nutritional disorders, poisonings, infections/infestations, skin/subcutaneous cellular tissue disorders, respiratory system disorders (Table 3). The *ICF* results for the 10 illness categories ranged from 0.75 to 0.95, with the highest for pain (*ICF* = 0.95; four species, 61 use reports), followed by muscular-skeletal system disorders (*ICF* = 0.94; four species, 53 use reports), digestive system disorders (*ICF* = 0.92; eight species, 87 use reports) and nutritional disorders (*ICF* = 0.91; three species, 23 use reports) (Table 3). Medicinal plants *Carum carvi* and *Hippophae tibetana* are most commonly used to treat pain. *C. carvi* and *H. tibetana* are also used to treat digestive system disorders, which are common in high-altitude areas, because of the high variety in diet or excessive consumption [35,37].

**Table 3.** Informant consensus factor for traditional medicinal plant use categories.

| Illness Category | Terms | Number of Taxa (Nt) | Number of Use Reports (Nur) | Informant Consensus Factor (ICF) |
|---|---|---|---|---|
| Pregnancy/Birth /Puerperium Disorders | Haemorrhages of pregnancy | 1 | 2 | - |
| Circulatory System Disorders | Hypertension | 1 | 18 | - |
| Pain | Toothache, Headache, Backache | 4 | 61 | 0.95 |
| Muscular-Skeletal System Disorders | Arthritis | 4 | 53 | 0.94 |
| Digestive System Disorders | Gastritis, Mountain sickness, Motion sickness, Diarrhoea, Biliousness | 8 | 87 | 0.92 |
| Nutritional Disorders | Restorative | 3 | 23 | 0.91 |
| Poisonings | Intoxication, Poisonings due to: bites and stings | 4 | 21 | 0.85 |
| lnfections/infest ations | Colds | 9 | 50 | 0.84 |
| Skin/Subcutaneo us Cellular Tissue Disorders | Nappy rash, Baldness, Acne, Heat rash, Demulcent, Emollient | 4 | 16 | 0.8 |
| Respiratory System Disorders | Pneumonia, Allergic rhinitis | 3 | 9 | 0.75 |

### 3.2.3. Wild Plants Used for Other Purposes

In addition to edible and medicinal plants, other categories were also documented in this study. Among them, there were 34 fodder plants, which had the largest number of species among all the use categories (Figure 6). Livestock and agriculture are the main livelihoods of Tibetans in Burang; therefore, many weeds (such as *Polypogon fugax*, *Pennisetum flaccidum* and *Clematis tangutica*) are used as fodder. After fodder plants, edible plants and medicinal plants, fuel (21 species) was the fourth most important category,

followed by incense plants (17 species). The winter in the Karnali valley, Burang is severely cold, with little snow. Plants such as *Salix sericocarpa*, *S. sclerophylla*, *Potentilla fruticosa* var. *pumila*, and *H. tibetana* are important heating materials for Tibetans in winter. Burning incense in rituals has always been an important religious practice, as Burang also has a strong religious atmosphere, burning incense is an important part of Tibetan daily life. *Nardostachys jatamansi*, *Rhododendron lepidotum*, *R. anthopogon*, *Artemisia roxburghiana*, *A. desertorum*, *Juniperus indica*, and *Waldheimia glabra* are important incense plants used here.

*Thymus linearis*, *Saussurea medusa*, and *S. tridactyla* are sold to add income. Three plants are used as dyes, the rhizomes of *Rheum moorcroftianum* and *Polygonum tortuosum* are used to dye the clothes of monks yellow, and the roots of *Arnebia euchr* are used to dye tsampa red for rituals. At the time of death and cremation, *Artemisia roxburghiana* is an essential plant to catch fire. The leaves of *P. tortuosum* are used as alternatives to tobacco plants.

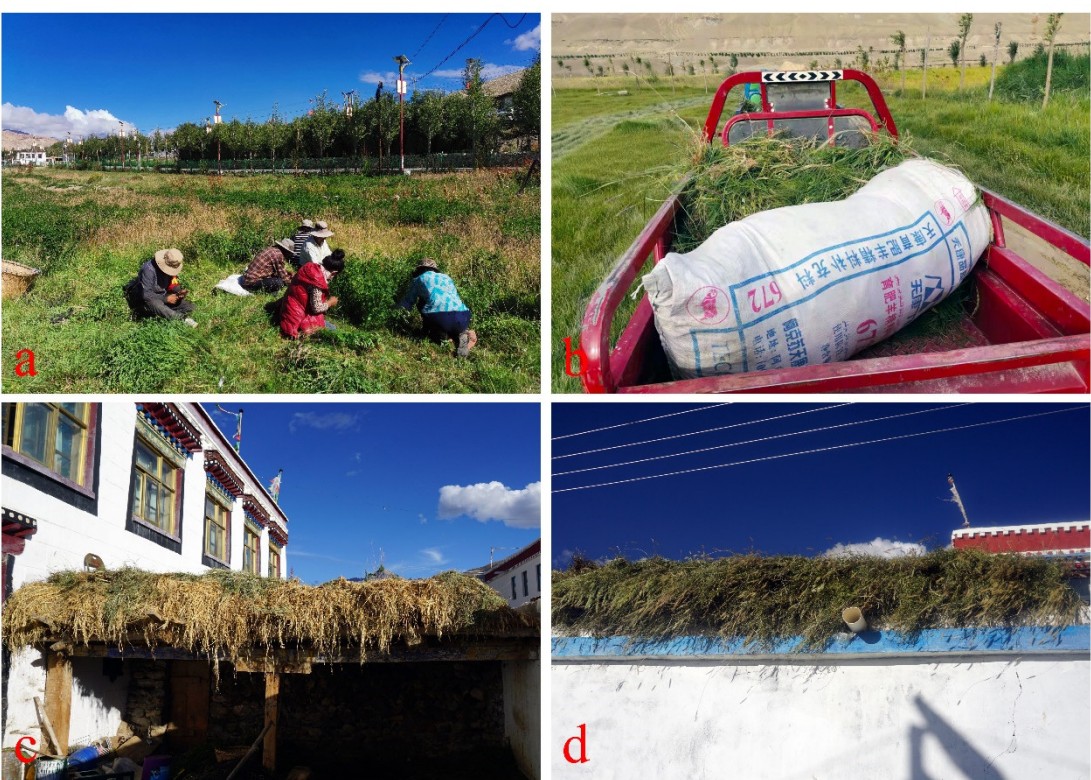

**Figure 6.** The process of fodder gathering. (**a**): Harvesting fodder; (**b**): shipping fodder; (**c**,**d**): drying fodder.

### 3.3. The Most Popularly Used Wild Plants

The *CI* was used to evaluate 76 plant species traditionally used by Tibetans, and the results show that the *CI* value ranges from 0.01–1.88. Based on the *CI* values (Table 3), the top five important plants in the Karnali River Valley are *Carum carvi* (*CI* = 1.88), *Hippophae tibetana* (*CI* = 1.45), *Rheum moorcroftianum* (*CI* = 0.87), *Urtica dioica* (*CI* = 0.78) and *Chenopodium album* (*CI* = 0.75) (Figure 7). These plants have been used by the Tibetans for a long time, and most of them still are.

These plant species are used for multiple purposes. For example, the fruit of *Hippophae tibetana* is a natural fruit that can be processed into vinegar or used to make decoctions to treat diseases; the leaves can be used as feed for yak; the stems are used as fuelwood in winter. From March to April, the tender leaves of *Carum carvi* are important local wild vegetables; ripe seeds are important seasonings, and can also be used to treat dentitis

and headaches. The petioles of *Rheum moorcroftianum* are peeled and eaten directly; the rhizomes can be applied to the joints to treat arthritis by soaking in wine or boiling water after drying; the rhizomes can also be used to dye yellow. The young leaves of *Urtica dioica* can be used as a vegetable or as a fodder; and the cooked young leaves can treat diarrhea and nourish the body. The use of young leaves of *Chenopodium album* is same as *Urtica dioica*. The difference is that the young leaves of *Chenopodium album*, leaves of *Allium przewalskianum*, and tsampa can be used to treat motion sickness when mixed.

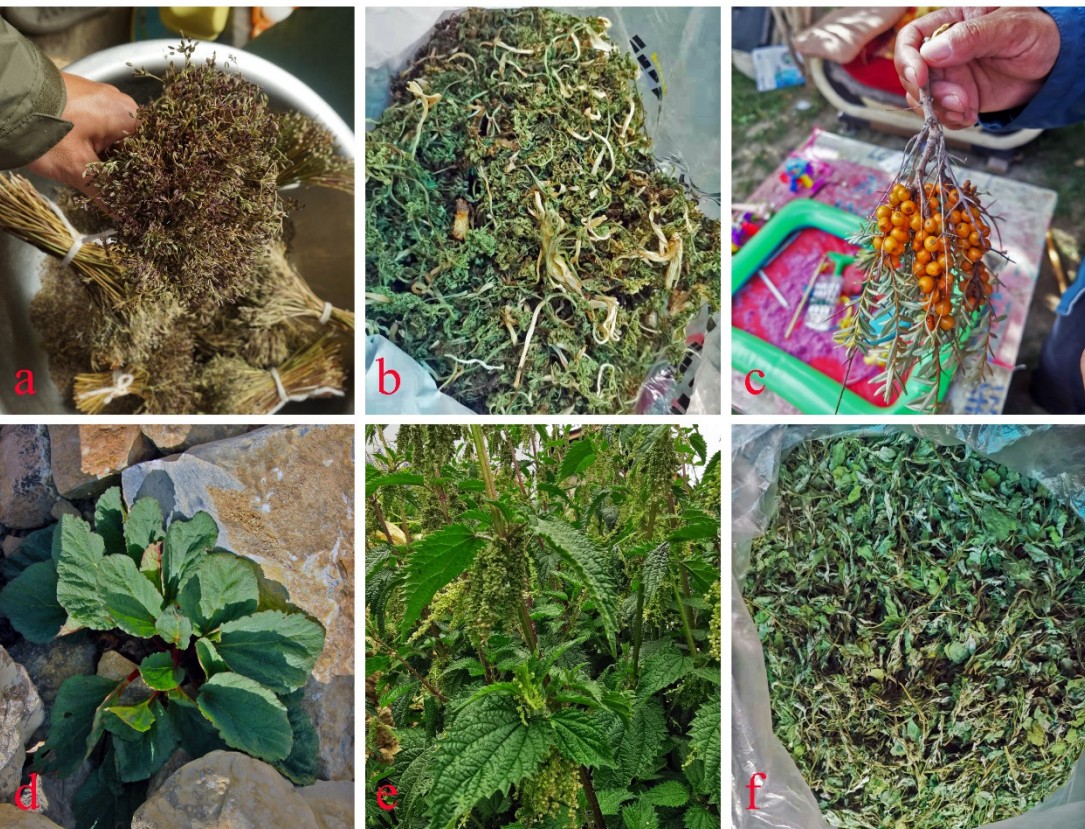

**Figure 7.** The top five wild plants used by Tibetans. (**a**): The seeds of *Carum carvi*; (**b**): the tender leaves of *Carum carvi*; (**c**): the fruits of *Hippophae tibetana*; (**d**): *Rheum moorcroftianum*; (**e**): *Urtica dioica*; (**f**): the tender leaves of *Chenopodium album*.

### 3.4. Comparison of Wild Edible Plants Used by Tibetans in Different Regions

To compare the homogeneity of wild edible plants utilised by Tibetans in different environments, we compared the edible plants used by Tibetans in eight regions (Burang, Shangri-la, Zhouqu, Zhagana, Sapi, Dhorpatan, Mustang and Lithang) of China [2,9,21,22]. Although traditional Tibetan medicinal plants in the Shangri-La region have been documented previously [23], the studies on other areas have mainly investigated edible plants; therefore, we did not compare the medicinal plant data for the sake of synergistic comparison. The results showed that the number of edible plants in the Karnali River Valley was less than that in the southern QTP (Figure 8). Only one plant *Chenopodium album* L., was found to be used by Tibetan people in seven regions (except for Dhorpatan). *Potentilla anserina* L. is used in Burang, Lithang, Zhagana, and Shangri-la. *Urtica dioica* L. is used in Burang, Zhagana, Mustang and Dhorpatan. *Carum carvi* L. is used in Burang, Zhouqu, Lithang and Mustang. *Rosa sericea* Wall. ex Lindl is used in Burang and Sapi. *Hippophae tibetana* Schltdl. is used in Burang and Mustang (Figure 8). The reasons for this low homogeneity of plants used by Tibetans in different regions may be related to different ecological environments. Several studies have also shown that extreme ecological conditions can affect lifestyle and culture [38–41]. This result is consistent with

previous research, and the use of medicinal plants by people living in environments with similar flora is also similar [42].

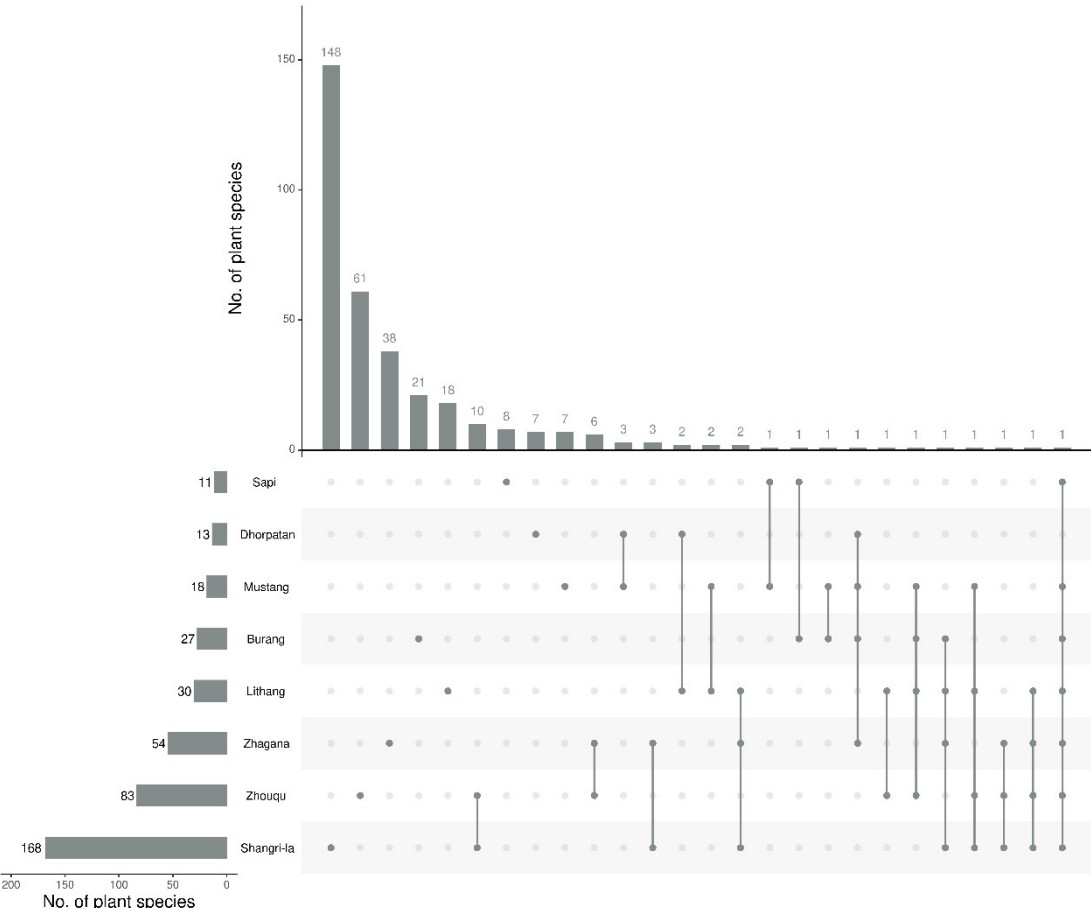

**Figure 8.** Comparison of plants used in eight regions. The bar chart on the left shows the original number of wild edible plants used by Tibetans in different regions; dot refers to the corresponding region name on the left in the intersection below; the connection between dots indicates that Tibetans in different regions jointly use some wild edible plants; upper histogram represents the number of plant species at this intersection.

## 4. Discussion

### 4.1. Important Wild Useful Plants

Based on the results of the *CI* quantitative analysis, we evaluated the top five wild plants that are important in the daily life of Tibetans in Burang Town (Figure 7). These plants have multiple uses as well as high recognition locally.

*Carum carvi* (*CI* = 1.88) is the most frequently reported wild plant in Burang Town. This plant is widely distributed in Europe, North America, North Africa, and Asia. It is also used by Tibetans in Zhouqu, Mustang, and Lithang. However, in Zhouqu, Mustang, and Lithang, only the seeds of *C. carvi* are used as a seasoning [2,9]. This may be related to the few vegetables grown in Burang Town in the past [27]. In Eurasia, the seed parts of *C. carvi* are commonly used for seasoning and medicinal consumption, such as in the Sakartvelo Republic of Georgia, Caucasus; Karakoram-Himalayan range, Pakistan; Udhampur, J&K, and in India [14,43,44]. It is also documented as a spice in Tibetan traditional medicine thangka [45]. However, few studies have documented the use of tender leaves. Considering the wide distribution of *C. carvi*, in-depth studies on the nutritional content of its tender leaves can be conducted in the future.

*Hippophae tibetana* (*CI* = 1.45) is mainly distributed in China (Gansu, Qinghai, Xizang), Bhutan, north India, and in Nepal at high altitudes (3600–4700 m). The fruits of *H. tibetana*

are eaten fresh or used to prepare a juice by Tibetans in the southern Mustang District, Nepal [2]. Many seabuckthorn plants (*Hippophae* sp.) are famous plants that represent the Tibetan culture. These have also been used in local diets, popular and learned Tibetan medicine, and today many seabuckthorn-based products have been developed by local companies. Tibetans living in Shangri-La, Yunnan, China, use the fruit of *Hippophae rhamnoides* L. subsp. *yunnanensis* Rousi as fresh fruit or use it to prepare beverages and wine [22]. Tibetans living in Zhouqu County, Gansu, China, consume the fresh fruit of *Hippophae rhamnoides* L. directly [9].

*Rheum moorcroftianum* (*CI* = 0.87), commonly known as rhubarb, is endemic to the Himalayas. In China, it is located in the western and central parts of Tibet and grows on sandy slopes or riverbank meadows at an altitude of 4500–5300 m. The rhizome are also one of the constituents of traditional Tibetan medicine, Chü-tsa [46]. In India, *R. moorcroftianum* is used to treat cold and internal injuries and is cultivated in some places [47–49]. The aqueous extracts of *R. emodi* and *R. moorcroftianum* rhizomes are widely used by the Bhotia tribe to dye wool, silk, and cotton fibers [50]. However, the wild conditions of *R. moorcroftianum* are worrying as they are disturbed by human collection [47,49,51].

*Urtica dioica* (*CI* = 0.78) is an important wild vegetable for Tibetans in Burang. The young leaves of *U. dioica* can be used as a vegetable or as fodder; the cooked young leaves can also be used to treat diarrhea and as nourishment. In traditional Tibetan culture, nettles are considered as food for hermits. *U. dioica* is also used as a wild vegetable and medicine in India [2].

*Chenopodium album* (*CI* = 0.75) is a widespread species [52]. It is consumed by Tibetans in Burang, Shangri-la, Zhouqu, Zhagana, and Lithang [2,9,21,22], and is also widely used worldwide [53–55]. It is thus a wild vegetable that can be cultivated.

### 4.2. The Relationship between Subsistence Patterns and Plant Utilisation in Burang

Our results show that Tibetans mainly collect wild plants from cropland and alpine pastures in the Kainali River Valley, Burang Town. Wild plants from alpine pastures were far less than those collected from croplands (Figure 3). This is not consistent with previous studies showing that as altitude increases, the number of useful plants decreases [56–58]. For example, with the exception of field vegetation, the species richness of all-use plants, edible plants, and medicinal plants used by the Lhoba decreased with increasing altitude [35]. Although there are fewer useful plants at higher altitudes than at lower altitudes, useful species (mainly medicinal plants) account for an average of 61% of the overall species richness, and the proportion of useful plants at different altitudes remains largely invariable [59].

Several hypotheses have been proposed to explore how and why people choose certain plants. For instance, the plant availability hypothesis states that certain plants that are more available and abundant locally are more likely to be used as medicines [60,61]. The physical distance from the home or community to the location of plant growth in the wild is often used as an indicator of availability [60,62]. Agriculture and animal husbandry are the main livelihoods of the Tibetans in Burang Town, and their daily lives revolve around these two practices. Therefore, it is not surprising that useful plants are collected in cropland and alpine pastures. In general, Tibetan herders stay in pastures for a long time and spend their daily lives in temporary shelters built on high mountain pastures. Therefore, many alpine plants have been identified and used. For example, *Polygonum affine* D. Don is used as a substitute grain and *Thymus linearis* Benth. is used as seasoning and medicine; *Potentilla fruticosa* var. *pumila* Hook.f. and *Caragana versicolor* Benth. are used as fuel plants, *Rhododendron anthopogon* D. Don and *R. lepidotum* Wall. ex G. Don are used as incense plants. In the cropland, we documented 53 useful plants, including 17 edible plants, 17 medicinal plants, 25 fodder plants, 11 fuel plants and 12 incense plants. Historically, only a small amount of *Brassica rapa* L. has been grown as a vegetable in Burang [27], and many local Tibetan vegetables are derived from cropland weeds. Thus, the number of useful plants from the cropland in Burang is far greater than that documented in other

parts of Tibet, especially forage plants and incense plants. In Burang, local women collect a large number of cropland weeds, dry them, and store them as winter fodder from August to September every year (Figure 6). This is consistent with previous studies showing that people are more inclined to collect useful plants close to their settlement or practice site based on the plant availability and accessibility [56,57]. Recent studies have shown that species diversity in agricultural systems close to forests is closely related to the ethnobotanical value of species [63]. The plant species in the agricultural system in Burang also exhibit high ethnobotanical values. And further ecological verification of species diversity is required in the future.

Human use of the natural environment is also affected by culture [25]. The use categories of wild plants as ritual, medicinal, and special uses, are influenced by cultural values [35]. Humanised natural environments are natural environments that have been transformed and utilised by humans to provide various products and services to humans under a specific culture. In Burang, the humanised natural environment (cropland) provides a living space for many plants, such as *Chenopodium album*, *Carum carvi*, *Hippophae tibetana*, and *Potentilla anserina*, which provide local people with various products and services. The rhizome of *Juncus thomsonii* Buchenau is consumed and is especially used by Tibetans in Burang. Weeds that grow around croplands are also an important source of fodder for livestock in winter. Ritual plants are also particularly important in daily Tibetan life [2]. Tibetans perform a cremation ceremony after death, and the material used to start the fire in cremation can only be *Artemisia roxburghiana* Bess in Burang. Further, Tibetans commonly have the custom of using incense, and a total of 17 incense plants were documented in Burang. Of these, 12 incense plants are collected from croplands, and four are collected from alpine pastures. *Artemisia desertorum* Spreng. is the best incense plant in the field whereas *Rhododendron anthopogon* D. Don is the best in the alpine pastures. The sacred mountain Kangrinboqe, considered the centre of the world, is located in Burang County. Every year, many believers worldwide visit to worship this sacred mountain, and Burang Town is an important transit point on the way. The famous Keja Temple (built in 996 AD) in Keja Village, Burang Town, is also visited every year by believers from China and Nepal [27]. The strong religious culture may explain why so many plants are used as incense in Burang Town.

Overall, our study shows that availability affects the collection location of useful plants and that local culture also affects plant use [25]. Further, many species in high-altitude agro-ecosystems have important ethnobotanical value, and need to be focused on in future research.

*4.3. Alpine Meadows and Agricultural Systems: An Important but Neglected Area of Plant Collection*

The environment of Qinghai-Tibet Plateau is one of the most vulnerable regions [64]. It is also densely populated, with a permanent population of 17.8326 million in 2020 [64]. Understanding how the residents of QTP use various plants can help understand their adaption to the plateau environment. This process is of great significance to the well-being of local people and protection of the ecological environment. The ethnobotanical survey in Burang gives a good example. The traditional agricultural system (cropland) of Tibetans provides a living environment for many wild plants, which provide Tibetans with products and services such as vegetables, fruits, fodder, and incense. Alpine meadows are also an important gathering site for local people to obtain vegetables, seasoning, fuelwood and incense plant species. Highland agricultural systems and alpine pastures thus preserve a wealth of traditional knowledge. For example, the Tajiks of the Pamirs determine the time and location of grazing based on the phenology of different species [65]. Based on previous research and this study, people living in the plateau have rich traditional knowledge regarding the uses of alpine plants. Thus, additional ethnobotanical investigations should be conducted on traditional cropland systems on the QTP in future, to better

understand the importance of traditional cropland systems to local people and the plateau ecological environment.

## 5. Conclusions

In this study, 76 wild plant species used by Tibetans were documented in the Karnali River Valley. These plants are used by Burang Tibetans to meet their daily needs of products and services (such as food and medicine). The number of edible plants in the Karnali River Valley is less than that in the southern QTP, but the structure of the plants used is similar, and there are few plants that are used commonly in different regions. The collection sites for these plants are mainly concentrated in the cropland and alpine pastures, which may have been previously neglected, especially cropland. The results of this study indicate that plant species collection sites can be influenced by the availability of plants and the local culture. Further, there are many species with important ethnobotanical values for locals in high-altitude agro-ecosystems. Future ethnobotanical studies are needed to examine agro-ecosystems in the plateau region in more detail. These findings will inform strategies and plans for local communities and governments to sustainably use and protect plants at high altitudes.

**Author Contributions:** Y.W. organized the study team and provided technical support and guidance. X.D. and C.G. designed and executed the research plan. X.D. wrote the manuscript. C.G., X.Z. and J.L. documented and organized the data. X.D. and C.G. identified the specimen and checked the information. All authors (included Y.J. and H.F.) took part in the field works. All authors were involved in the drafting and revision of the manuscript and approved the final revision. All authors have read and agreed to the published version of the manuscript.

**Funding:** The study was funded by Ministry of Science and Technology of the People's Republic of China "the Second Tibetan Plateau Scientific Expedition and Research (No. 2019QZKK0502)".

**Institutional Review Board Statement:** The authors asked for permission from the local authorities and the people interviewed to carry out the study.

**Informed Consent Statement:** The people interviewed were informed about the study's objectives and the eventual publication of the information gathered, and they were assured that the informants' identities would remain undisclosed.

**Data Availability Statement:** Please contact the corresponding author for data requests.

**Acknowledgments:** We are very grateful to the informants for sharing their knowledge with us. We thank Pei Shengji for technical guidance. In addition, we thank Xu Haikun as auto drivers in the wild works.

**Conflicts of Interest:** The authors declare that they have no competing interests.

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
