# Peer review of "Wild Plants Used by Tibetans in Burang Town, Characterized by Alpine Desert Meadow, in Southwestern Tibet, China"

_agronomy, doi:10.3390/agronomy12030704_

Round 1
Reviewer 1 Report
The authors do in fact present a very well researched and written manuscript.
There are some small corrections needed:
Figure 3: for the color bars it should be "plants" rather than "plant"
Figure 5: Scientific names must be italicised
Figure 7: What ARE the plants? There are no species given in the legend
Otherwise the papers is well prepared, and will be an interesting contribution to ethnobotanical studies in the Himalayas.
Author Response
Thank you for your hard work, please see the attachment for your comments.Since we did not find a place to upload manuscripts and revision letters,
we will upload them in time after asking the editor.

Reviewer 2 Report
Indeed field studies in Xizang Tibet are scarced and I am happy about the progress. The paper is well-structured, well-written with good English. I highly appreciate providing Tibetan script version. Well done! The paper can be accepted after minor changes. Please check for typos: p. 14 Waldheimia Glabra > Waldheimia glabra p. 15 syudies > studies I recommend also mentioning this paper: Lognay, G., Haubruge, E., Wathelet, B., Mathieu, F., Marlier, M., Malaisse, F., 2008. Ophioglossum polyphyllum A. Braun in Seub. (Ophioglossaceae, Pteridophyta), a rare potherb in south central Tibet (TAR, PR China). Geo-Eco-Trop 32:9-16. and maybe this about the importance of agriecosystems for world food systems: Turner, N.J., Łuczaj, Ł.J., Migliorini, P., Pieroni, A., Dreon, A.L., Sacchetti, L.E. and Paoletti, M.G., 2011. Edible and tended wild plants, traditional ecological knowledge and agroecology. Critical Reviews in Plant Sciences, 30(1-2), pp.198-225. Maybe typical dishes of wild food plants can be characterized in a few sentences?Author Response
Thank you for your hard work, please see the attachment for your comments.Since we did not find a place to upload manuscripts and revision letters,
we will upload them in time after asking the editor.
